# Unlearning Personal Data from a Single Image

**Thomas De Min**[†]
*University of Trento, Italy*
*thomas.demin@unitn.it*

**Massimiliano Mancini**
*University of Trento, Italy*

**Stéphane Lathuilière**
*LTCI, Télécom-Paris, Institut Polytechnique de Paris, France*
*Inria Grenoble, Univ. Grenoble Alpes, France*

**Subhankar Roy**
*University of Trento, Italy*

**Elisa Ricci**
*University of Trento, Italy*
*Fondazione Bruno Kessler, Italy*

**Reviewed on OpenReview:** *https://openreview.net/forum?id=VxC4PZ71Ym*

## Abstract

Machine unlearning aims to erase data from a model as if the latter never saw them during training. While existing approaches unlearn information from complete or partial access to the training data, this access can be limited over time due to privacy regulations. Currently, no setting or benchmark exists to probe the effectiveness of unlearning methods in such scenarios. To fill this gap, we propose a novel task we call One-Shot Unlearning of Personal Identities (1-SHUI) that evaluates unlearning models when the training data is not available. We focus on unlearning identity data, which is specifically relevant due to current regulations requiring personal data deletion after training. To cope with data absence, we expect users to provide a portraiting picture to aid unlearning. We design requests on CelebA, CelebA-HQ, and MUFAC with different unlearning set sizes to evaluate applicable methods in 1-SHUI. Moreover, we propose MetaUnlearn, an effective method that meta-learns to forget identities from a single image. Our findings indicate that existing approaches struggle when data availability is limited, especially when there is a dissimilarity between the provided samples and the training data. Source code available at `github.com/tdemin16/one-shui`.

## 1 Introduction

In the sci-fi movie franchise Men in Black, the secret service agents possess a Neuralyzer, a brainwashing device that, with a bright flash, selectively wipes out anyone's memory of alien encounters. Not being limited to fiction, a "Neuralyzer"-like tool is also practical for machine learning practitioners in order to erase information from trained neural networks. Such a requirement can inevitably arise when data owners exercise their "right to be forgotten" (Voigt & Von dem Bussche, 2017; Mantelero, 2013; Goldman, 2020). Given the practical implications, Machine Unlearning (MU) aims to forget the influence of *targeted information* from a trained model, as if they were not part of the training set.

Some approaches are designed to forget a class or a subclass from the model (Chundawat et al., 2023a; Chen et al., 2023). However, more realistic methods are conceived to unlearn a random subset of the

---

[†]Corresponding Author.

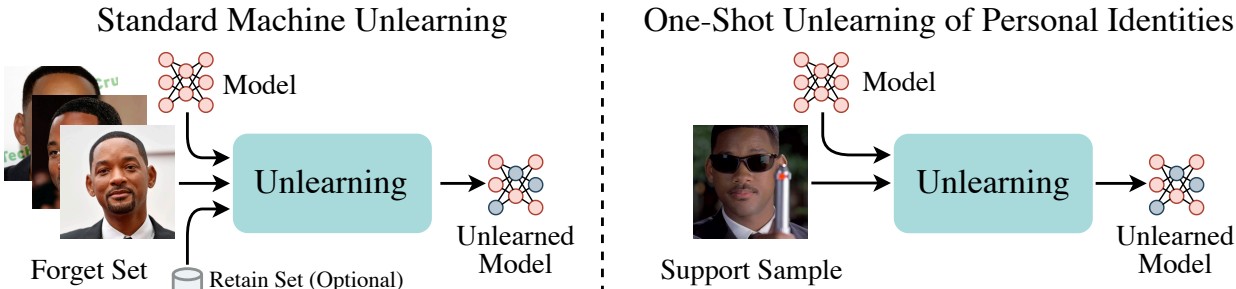

Figure 1: **One-Shot Unlearning of Personal Identities.** Standard machine unlearning approaches leverage the entire forget set to forget an identity. In 1-SHUI, the user provides a picture of themselves as the only input to the unlearning algorithm for forgetting their identity. 1-SHUI evaluates unlearning when the entire training set is inaccessible.

training data (Jia et al., 2023; Fan et al., 2024; Cheng & Amiri, 2023; Shen et al., 2024; He et al., 2024; Zhao et al., 2024b; Huang et al., 2024). These works require full (Fan et al., 2024; Jia et al., 2023) or partial (Foster et al., 2024b; Hoang et al., 2024) access to the training data as they compute gradient updates with carefully designed loss functions or identify important parameters for the information to unlearn. Despite their promising results, assuming access to the original dataset is unrealistic under current data protection regulations (Mantelero, 2013; Goldman, 2020), e.g. Article 6 of EU Directive 95/46/EC forbids to store personal data "for no longer than is necessary". To address this challenge, existing works (Chundawat et al., 2023b; Kravets & Namboodiri, 2024; Tarun et al., 2023) propose to exploit reconstruction-based algorithms to aid unlearning of categories without accessing the original data. However, these techniques can only be applied to class-unlearning scenarios, not being able to reconstruct specific data points. Thus, they cannot be used for targeted unlearning (i.e., random data unlearning), limiting their applicability.

The objective of this work is to allow for unlearning randomized samples in data absence scenarios. To achieve this, we propose a novel task, One-Shot Unlearning of Personal Identities (1-SHUI), which evaluates unlearning when training data is unavailable. 1-SHUI focuses on identity unlearning, a realistic scenario that accounts for unlearning personal data to protect privacy, which is a special case of random unlearning. The target of 1-SHUI are machine learning models trained for an arbitrary downstream task (e.g., face attribute recognition, age classification), performing unlearning at the identity level. To circumvent the lack of training data, we propose that unlearning requests from a user are accompanied by a single "Support Sample" portraying their identity to the unlearning algorithm. Therefore, unlearning methods must unlearn the whole data associated with the identity without accessing the real forget data, but to this single sample.

Since unlearning using a single image per identity is challenging, we propose a learning-to-unlearn framework, MetaUnlearn, that trains a meta-loss function to approximate an unlearning loss without needing the full training data. During training, MetaUnlearn exploits the available training data (before their removal) to simulate unlearning requests. By sampling random identities and their respective Support Sets, MetaUnlearn iteratively exploits the Support Set to unlearn IDs while using the rest of the data to compute the error estimate, which is backpropagated to optimize MetaUnlearn parameters. Although 1-SHUI is challenging, MetaUnlearn achieves consistent forgetting results compared to existing approaches across different datasets (i.e., CelebA (Liu et al., 2015), CelebA-HQ (Karras et al., 2017; Lee et al., 2020), MUFAC (Choi & Na, 2023)) and unlearning set sizes.

**Contributions.** In summary, our contributions are the following:

1. We are the first to highlight the mismatch between realistic data availability and existing unlearning evaluation benchmarks.

2. We propose 1-SHUI, the first benchmark for methods that perform targeted unlearning in absence of training data.

3. We evaluate existing applicable MU methods on 1-SHUI, revealing that they struggle to tackle machine unlearning when training data is unavailable.

4. We propose MetaUnlearn the first approach tailored for unlearning in data absence, showing its effectiveness in 1-SHUI.

## 2 Related Work

**Machine Unlearning.** By retraining (part of) the network weights, exact machine unlearning methods (Aldaghri et al., 2021; Bourtoule et al., 2021; Yan et al., 2022) can guarantee that training data is completely forgotten. However, retraining even part of the model makes these approaches prohibitively expensive (Chundawat et al., 2023a; Foster et al., 2024b; Graves et al., 2021). As an efficient alternative, approximate unlearning algorithms (Chundawat et al., 2023a; Trippa et al., 2024; Choi & Na, 2023; Fan et al., 2024; Foster et al., 2024b;a; Chen et al., 2023; Chundawat et al., 2023b; Jia et al., 2023) propose to remove sample influence by tuning model parameters for just a few learning steps, relaxing guarantee constraints. Most approximate unlearning works are designed for class (Chen et al., 2023; Chundawat et al., 2023a; Cheng et al., 2024; Hoang et al., 2024; Zhao et al., 2024a) or random (Jia et al., 2023; Fan et al., 2024; Kurmanji et al., 2023; Cheng & Amiri, 2023; He et al., 2024; Huang et al., 2024; Shen et al., 2024) unlearning, thus, they either focus on forgetting a single class or a random subset of the training dataset. However, both these lines of work assume full or partial access to the training dataset, which can be limited in time (Mantelero, 2013). To circumvent data requirements, Chundawat et al. (2023b) proposes reconstructing forget information via an error maximization noise generation process (Tarun et al., 2023) aiming to invert the learning process. In the same paper, Chundawat et al. (2023b) proposes another method that exploits knowledge distillation (Hinton, 2014) and a generator network to distill synthetic information from the original model to a learning student while bypassing forget-related information. Instead, Kravets & Namboodiri (2024) proposes to unlearn classes from CLIP (Rombach et al., 2022) by generating synthetic samples via gradient ascent (Szegedy, 2013).

Although these methods demonstrate unlearning capabilities in a data-absence scenario, they are limited by reconstruction-based algorithms, which use the model **output** distribution as the only source of information about training data. Therefore, they cannot unlearn information that is not linked to the output (e.g., random data, identity data). To circumvent this limitation, we propose a novel task called One-Shot Unlearning of Personal Identities. By requesting users for a picture portraying themselves, 1-SHUI exploits the semantic similarity of images of the same identity to overcome the data absence constraint. The requested image aids unlearning and is discarded after use.

**Meta-Learning.** Contrary to standard machine learning, meta-learning distills knowledge from multiple learning episodes for improved learning performance, a paradigm known as learning to learn (Hospedales et al., 2021). Preliminary meta-learning approaches (Finn et al., 2017; Nichol et al., 2018; Antoniou et al., 2018) focused on learning the model initialization to facilitate few-shot learning, via e.g. first (Nichol et al., 2018) or second-order (Finn et al., 2017), or specific training recipes (Antoniou et al., 2018). Beyond few-shot learning, meta-learning has been applied to other tasks, such as domain generalization (Li et al., 2019), continual learning (Javed & White, 2019), and fast model adaptation (Bechtle et al., 2021). Related to our work, Huang et al. (2024) achieves a better forgetting and retaining balance via a meta-learning-based approach that improves over existing baselines. Gao et al. (2024) proposes a meta-learning framework to mitigate the degradation of benign concepts that are related to unlearned (harmful) ones. However, both works cannot be used in data absence scenarios as both require original data availability.

Following meta-learning principles, our proposed approach MetaUnlearn approximates the unlearning process by simulating multiple unlearning requests. At the end of the training, MetaUnlearn can generalize unlearning by accessing only a single sample per identity.

# 3 Problem Formulation

This section introduces the machine unlearning problem and the proposed One-Shot Unlearning of Personal Identities task, highlighting how the benchmark dataset is constructed in practice. Section 3.1 provides a machine unlearning overview, focusing on identity unlearning. Section 3.2 introduces the novel 1-SHUI task, and describes its relevance in realistic scenarios. Finally, section 3.3 contains a schematic overview of how we split the training data to construct the benchmark dataset.

## 3.1 Machine Unlearning

Let $f_\theta : \mathcal{X} \to \mathcal{Y}$ be a model parameterized by $\theta$ that maps inputs from the image domain $\mathcal{X}$ to the target domain $\mathcal{Y}$ (e.g., face attributes), with $\theta$ trained on dataset $\mathcal{D}_{tr}$. The dataset is characterized by image-label pairs such that $\mathcal{D}_{tr} = \{(\mathbf{x}_j, y_j)\}_{j=1}^N$, with $N$ the dataset size. The goal of machine unlearning is to "forget" a subset of the training dataset $\mathcal{D}_f \subset \mathcal{D}_{tr}$ while preserving original model performance on the retain $\mathcal{D}_r = \mathcal{D}_{tr} \setminus \mathcal{D}_f$ and test $\mathcal{D}_{te}$ datasets. Given original model parameters $\theta$ and a forget set, an effective MU algorithm $\mathcal{U}$ outputs model weights $\theta_u$ that have minimal distance w.r.t. the optimal parameters $\theta_r$ computed by retraining the model $f(\cdot)$ on $\mathcal{D}_r$:

$$\theta_u = \underset{\theta_u}{\arg\min}\, d(\theta_u, \theta_r), \tag{1}$$

where $d$ is a distance metric measuring the closeness between the two model ($\theta_u$ and $\theta_r$) distributions (Zhao et al., 2024b).

While in random sample unlearning (Foster et al., 2024b; Golatkar et al., 2021) the forget set samples are i.i.d., in the identity unlearning scenario (Choi & Na, 2023) samples of the same identity correlate. Let us denote as $\mathcal{I}_{tr}$ the training, $\mathcal{I}_f$ the forget, and $\mathcal{I}_r$ the retain identities, with $\mathcal{I}_f = \mathcal{I}_{tr} \setminus \mathcal{I}_r$. We construct $\mathcal{D}_f$ from $\mathcal{I}_f$:

$$\mathcal{D}_f = \{(\mathbf{x}_j, y_j, i_j) \mid i_j \in \mathcal{I}_f\}_{j=1}^{N_f}, \tag{2}$$

where $i_j$ is the identity number, and $N_f$ is the forget set size.

## 3.2 One-Shot Unlearning of Personal Identities

As section 1 shows, complete or partial access to the original dataset is not guaranteed in realistic scenarios (Mantelero, 2013; Goldman, 2020). To circumvent this limitation, this work proposes One-Shot Unlearning of Personal Identities (1-SHUI) a novel benchmark that requires unlearning methods to operate without access to training data at unlearning time. Specifically, we analyze the case of identity unlearning where the entire training dataset is discarded after training, as it might contain sensitive data. One-Shot Unlearning of Personal Identities exploits the semantic similarity of images of the same identity to cope with the challenges of this setting. In practice, we require users asking to be unlearned to provide one image with their identity portrayed. We name this image the *Support Sample*. This sample aids unlearning and is discarded after use. We define the collection of Support Samples from the same unlearning request as:

$$\mathcal{S} = \{(\mathbf{x}_j, y_j, i_j) \mid i_j \in \mathcal{I}_f \wedge (\forall k \in \{1, ..., N_\mathcal{S}\}, i_k = i_j \iff k = j)\}_{j=1}^{N_\mathcal{S}}. \tag{3}$$

Thus, the Support Set contains only one data point for each identity, where $N_\mathcal{S}$ is the number of identities to unlearn. Given the Support Set $\mathcal{S}$ and the original model weights $\theta$, the unlearning algorithm outputs $\theta_u$ to minimize equation 1: $\mathcal{U}(\theta; \mathcal{S}) = \theta_u$. Therefore, $\mathcal{U}$ must unlearn $\mathcal{D}_f$ by having only access to $\mathcal{S}$. This constraint makes the task extremely challenging as (i) methods cannot access the retain or the forget sets, and (ii) the Support Sample cannot capture the entire distribution of the forget samples we aim to remove.

## 3.3 Benchmark Construction

To evaluate methods in 1-SHUI, we split the full dataset $\mathcal{D}$ in train $\mathcal{D}_{tr}$, validation $\mathcal{D}_v$, and test $\mathcal{D}_{te}$ sets with non-overlapping identities, as figure 2 illustrates. Then, we randomly select $N_\mathcal{S}$ identities from $\mathcal{D}_{tr}$

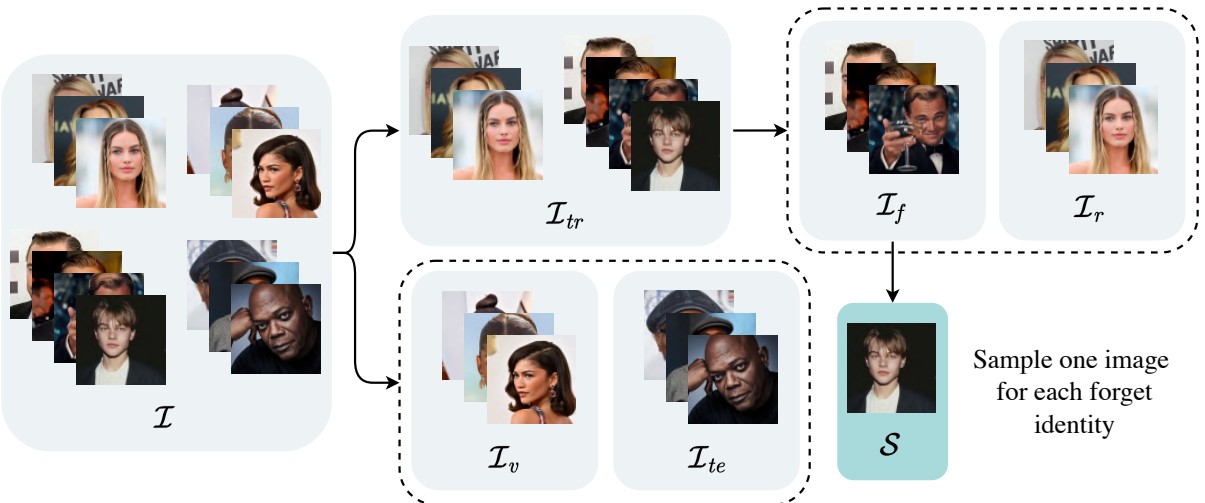

Figure 2: **Benchmark dataset construction.** The dataset is split based on identities, dividing them into train $\mathcal{I}_{tr}$, validation $\mathcal{I}_v$, and test $\mathcal{I}_{te}$ IDs. Following, we sample forgetting identities from training ones, splitting $\mathcal{I}_{tr}$ into forget $\mathcal{I}_f$ and retain $\mathcal{I}_r$ IDs. Out of forget identities, we sample one image for each identity to form the Support Set $\mathcal{S}$, which is unavailable at training time.

that serve as the base to construct the forget dataset $\mathcal{D}_f$. The remaining IDs are left for the retain set $\mathcal{D}_r$. For each identity in the forget set, we randomly remove one image to form the Support Set $\mathcal{S}$, such that the same image cannot appear in both $\mathcal{D}_f$ and $\mathcal{S}$. We note that the Support Set is also removed from the training dataset, and thus unseen by the model: $\mathcal{S} \cap \mathcal{D}_{tr} = \varnothing$, and consequently $\mathcal{S} \cap \mathcal{D}_f = \varnothing$. After splitting the dataset, we train the model on $\mathcal{D}_{tr}$ and use $\mathcal{S}$ to unlearn $\mathcal{D}_f$. Instead, the evaluation is performed on $\mathcal{D}_r$, $\mathcal{D}_f$, and $\mathcal{D}_{te}$.

# 4   Learning to Unlearn for 1-SHUI

This section presents MetaUnlearn, our proposed approach tailored for 1-SHUI that unlearns identities using Support Samples only. MetaUnlearn exploits the dataset to learn an unlearning loss so that it can forget identities through the limited set of Support Samples, before discrding the training data. Thus, our proposed approach consists of three stages: (i) model training on the downstream task; (ii) learning of the unlearning loss; (iii) unlearning via the trained loss when data is not available anymore. Figure 3 summarizes these three steps intuively. Section 4.1 describes how we train MetaUnlearn in a tractable and scalable way to generalize unlearning without accessing original data. Section 4.2, instead, presents how the trained MetaUnlearn can be used to unlearn in data absence.

## 4.1   MetaUnlearn training

The learnable loss $h_\phi(\cdot)$ (step ii) requires simulating unlearning requests to estimate $h_\phi(\cdot)$ unlearning error and optimize its parameters. Before the data is discarded, simulated unlearning requests are formed by randomly sampling $N_\mathcal{S}$ identities from the training set and split $\mathcal{D}_{tr}$ following the same procedure of section 3.3. We, therefore, forward simulated support samples through the original network and the meta-loss and compute a gradient step w.r.t. model's parameters $\theta$ to produce unlearned parameters $\theta_u$. We evaluate the effectiveness of unlearning and backpropagate the error to $h_\phi(\cdot)$ parameters. We discard $\theta_u$, simulate another unlearning request, and iterate until convergence (further details in appendix A). $N_\mathcal{S}$ is kept fixed throughout training and, to ensure all identities are used, ID sampling is done without replacement.

Following previous works in meta-learning (Li et al., 2019; Bechtle et al., 2021), we design the learnable loss $h_\phi(\cdot)$ as an MLP parameterized by $\phi$. We aim to train the meta-loss in such a way that computing an optimization step using the gradient of the meta-loss $\nabla_\theta h_\phi(f_\theta(\mathcal{S}))$ minimizes equation 1. Therefore, we

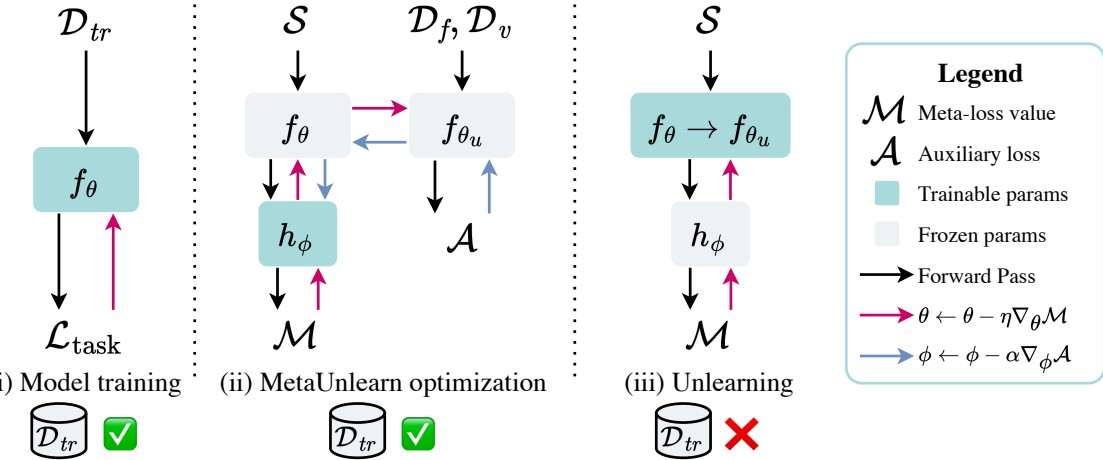

Figure 3: **MetaUnlearn pipeline.** While training data are available (i) The original model $f_\theta$ is trained on $\mathcal{D}_{tr}$ using the task loss function (e.g., cross-entropy loss). (ii) Before being discarded, the training data can be used to learn the proposed unlearning loss (MetaUnlearn). MetaUnlearn simulates an unlearning request $\mathcal{S}$ and use the meta-loss ($h_\phi(\cdot)$) to unlearn it at once. We evaluate the unlearned model $f_{\theta_u}$ using equation 6 ($\mathcal{A}$) on the forget, and validation data and backpropagate to $h_\phi(\cdot)$. $f_{\theta_u}$ is discarded, we simulate another unlearning request and iterate until convergence. (iii) Once the MetaUnlearn is trained, we can use it to unlearn identities via Support Samples, without accessing the original training set.

iteratively simulate unlearning requests and optimize $h_\phi(\cdot)$ parameters with:

$$\phi = \arg\min_\phi d\left(\theta - \eta\nabla_\theta h_\phi(f_\theta(\mathcal{S})), \theta_r\right), \tag{4}$$

where $\eta$ is the learning rate and $\nabla_\theta$ computes the gradient of the meta-loss w.r.t. the model parameters. Compared to equation 1, the minimization is now carried through the learnable parameters $\phi$ and not $\theta_u$.

Although directly optimizing equation 4 is computationally tractable, its minimization is not scalable as it requires computing retrained parameters $\theta_r$ for each simulated request. Therefore, we propose to evaluate the unlearning effectiveness by exploiting an auxiliary error function $\mathcal{A}$ that is agnostic to $\theta_r$. However, computing a reliable error estimate without the retrained weights is challenging. Assuming that the unseen sample losses for retrained and original models are similar for sufficiently small forget sets, we propose that $\mathcal{A}$ minimizes the discrepancy between the average forget loss and the original model validation loss. Intuitively, forget loss values should be close to the loss of unseen samples of the retrained model, on average. Thus, we optimize $\phi$ as follows:

$$\phi = \arg\min_\phi \mathcal{A} = \arg\min_\phi \underbrace{\|\mathcal{L}_{\text{task}}(\mathcal{D}_f; \theta_u) - \mathcal{L}_{\text{task}}(\mathcal{D}_v; \theta_u)\|_2^2}_{\text{Forget and unlearning test loss alignment}} + \underbrace{\|\mathcal{L}_{\text{task}}(\mathcal{D}_f; \theta_u) - \mathcal{L}_{\text{task}}(\mathcal{D}_v; \theta)\|_2^2}_{\text{Forget and original test loss alignment}}, \tag{5}$$

where $\theta_u = \theta - \eta\nabla_\theta h_\phi(f_\theta(\mathcal{S}))$, and $\mathcal{L}_{\text{task}}$ is the loss used to train the model $f_\theta$ on the downstream task (e.g., cross-entropy loss). The first term of equation 5 aligns forget and validation sets loss of the unlearned model to ensure they are close. The second term aligns the forget loss of the unlearned and original model, preserving performance. Together, the two terms push forget data to be treated as unseen (i.e., validation) w.r.t. both its current status (first term) and the original model (second term). We note that the gradient $\nabla_\theta h_\phi(f_\theta(\mathcal{S}))$ is computed at once for all Support Samples in $\mathcal{S}$. Therefore, equation 5 evaluates unlearning after just one gradient update.

Although equation 5 aligns forget and validation set losses, we noticed that, in some cases, this did not correspond to an alignment also in accuracy terms. To explicitly account for mismatches in the accuracy between forget and validation set, we scale each loss term in equation 5 by the opposite of its corresponding

accuracy:

$$\phi = \arg\min_{\phi} \mathcal{A} = \arg\min_{\phi} \|(1 - \mathrm{mAP}(\mathcal{D}_f; \theta_u)) \cdot \mathcal{L}_{\mathrm{task}}(\mathcal{D}_f; \theta_u) - (1 - \mathrm{mAP}(\mathcal{D}_v; \theta_u)) \cdot \mathcal{L}_{\mathrm{task}}(\mathcal{D}_v; \theta_u)\|_2^2 +$$
$$+ \|(1 - \mathrm{mAP}(\mathcal{D}_f; \theta_u)) \cdot \mathcal{L}_{\mathrm{task}}(\mathcal{D}_f; \theta_u) - (1 - \mathrm{mAP}(\mathcal{D}_v; \theta)) \cdot \mathcal{L}_{\mathrm{task}}(\mathcal{D}_v; \theta)\|_2^2. \tag{6}$$

To summarize, the first term of the auxiliary loss promotes similar loss values for the unlearned model between forget and unseen samples, making them less distinguishable. Then, the second term ensures that original model loss values for unseen samples are preserved, limiting performance degradation. Finally, scaling by the opposite of the accuracy improves accuracy alignment between forget and test sets. Once the meta-loss is trained, the dataset can be discarded.

## 4.2 MetaUnlearn unlearning

By simulating unlearning requests multiple times, the meta-loss learns to forget all identities in the training dataset by only relying on Support Samples $\mathcal{S}$. In order to forget identities at unlearning time (step (iii)), it is sufficient to forward $\mathcal{S}$ through $h_\phi(f_\theta(\cdot))$ and update $\theta$ with a single gradient step: $\theta_u \leftarrow \theta - \eta\nabla_\theta h_\phi(f_\theta(\mathcal{S}))$. Similarly to equation 5 and 6, all identities are unlearned at once via one gradient update. Algorithm 1 summarizes the training and unlearning procedures (step (ii) and (iii)) of MetaUnlearn.

# 5 Experiments

Section 5.1 describes 1-SHUI experimental setting, outlining datasets, metrics, and baselines used (see section A for implementation details). Section 5.2 evaluates existing baseline approaches and our MetaUnlearn in 1-SHUI, while section 5.3 compares methods when the number of unlearning requests is extremely low. Instead, sections 5.4 and 5.5 investigate when unlearning is the hardest and when unlearning harms model performance. Finally, section 5.6 provides a complete ablation of MetaUnlearn.

## 5.1 Experimental setting

**Datasets.** To evaluate unlearning approaches in 1-SHUI, we focus on datasets annotated with identity information and for a different downstream task (i.e., face attribute recognition, age classification). The goal is to train a model on the downstream task and to perform identity-aware unlearning while preserving the original model accuracy on the test set. We identify three datasets that satisfy our requirements: CelebA-HQ (Karras et al., 2017; Lee et al., 2020), CelebA (Liu et al., 2015), and MUFAC (Choi & Na, 2023). The first two datasets contain images of celebrities annotated with 40 facial attributes

---

**Algorithm 1** MetaUnlearn pseudocode.

```
def simulate_unlearning(I_tr, D_tr):
  I_f ~ I_tr  # sampling w/o replacement
  D_f = {(x_j, y_j, i_j) | i_j ∈ I_f}_{j=1}^{N_f}
  D_r = D_tr \ D_f
  S = build_support_set(I_f, D_f)
  return S, D_f, D_r

def MetaUnlearn_training(I_tr, D_tr, D_v):
  for epoch in range(num_epochs):
    # iterate all IDs in batches of size N_S
    for it in range(⌈N/N_S⌉)
      # compute simulated unlearning step
      S, D_f, D_r = simulate_unlearning(I_tr, D_tr)
      M = h_φ(f_θ(S))
      θ_u = θ - η∇_θ M  # unlearn S in one step
      # evaluate meta-loss
      A = MSE(L_task(D_f; θ_u), L_task(D_v; θ_u))
      A += MSE(L_task(D_f; θ_u), L_task(D_v; θ))
      # update φ using gradients ∇_φ, and lr α
      φ = Adam(φ, ∇_φ A, α)
  return None

def MetaUnlearn_unlearning(S):
  M = h_φ(f_θ(S))
  θ = θ - η∇_θ M  # unlearn S in one step
  return θ
```

---

and identity information. The third, instead, is built from family images, where each identity is portrayed multiple times with photos spanning different ages (e.g., from newborn to teenager), and is annotated for eight different age categories. We investigate two unlearning sizes: 20 and 50 identities for CelebA-HQ,

namely CelebA-HQ/20 and CelebA-HQ/50, and 5 and 10 IDs for CelebA and MUFAC, i.e. CelebA/5, CelebA/10, MUFAC/5, and MUFAC/10. We set a lower number of identities for CelebA and MUFAC, as each counts approximately 20 and 10 samples/identity, respectively, compared to CelebA-HQ's 5 samples/identity.

**Metrics.** We report retain, forget, and test accuracy, for each method to estimate the unlearning effectiveness. As ranking approaches over three different metrics is challenging, we summarize the three reported accuracies using the Tug of War (ToW) metric (Zhao et al., 2024b). ToW captures the performance discrepancy with the retrained model on the three datasets in a single score, simplifying evaluation:

$$
\begin{aligned}
\text{ToW} = & \left[1 - |\text{mAP}(\mathcal{D}_r; \theta_u) - \text{mAP}(\mathcal{D}_r; \theta_r)|\right] \cdot \\
& \cdot \left[1 - |\text{mAP}(\mathcal{D}_f; \theta_u) - \text{mAP}(\mathcal{D}_f; \theta_r)|\right] \cdot \\
& \cdot \left[1 - |\text{mAP}(\mathcal{D}_{te}; \theta_u) - \text{mAP}(\mathcal{D}_{te}; \theta_r)|\right].
\end{aligned}
\tag{7}
$$

Like previous approaches (Kurmanji et al., 2023; Hoang et al., 2024), we additionally report the membership inference attack (MIA) (Shokri et al., 2017; Carlini et al., 2022; Yeom et al., 2018; Zarifzadeh et al., 2024), measuring the true positive rate in inferring sample membership at 0.01% false positive rate (further details in section B). We denote in **bold** the best method and underline the second best on each metric.

**Baselines.** Although several machine unlearning approaches exist, only a handful can be applied to 1-SHUI. Methods that only assume access to the forget data are the only applicable algorithms for our setting, as we can carry unlearning via Support Samples only. Thus, we compare MetaUnlearn with SSD (Foster et al., 2024b), PGU (Hoang et al., 2024), and JiT (Foster et al., 2024a). While SSD unlearns via parameter dampening, PGU and JiT unlearn the model via gradient updates. Note that not performing meta learning and replacing $h_\phi(\cdot)$ with the losses proposed in Hoang et al. (2024) or Foster et al. (2024a), would be equivalent to using PGU, and JiT respectively, with the only difference that MetaUnlearn forgets all identities in a single gradient update. Thus the comparisons with these methods already validate the effectiveness of our meta-learning approach. We also report SCRUB (Kurmanji et al., 2023) and Bad Teacher (Chundawat et al., 2023a) that perform unlearning using the entire training dataset: their performance is used as a reference.

## 5.2 Results in One-Shot Unlearning of Personal Identities

Tables 1 to 3 evaluate existing methods and MetaUnlearn in One-Shot Unlearning of Personal Identities. We divide both tables into two sections corresponding to different unlearning set sizes. From top to bottom, we report pretrain and retrained model performance for each size, as previous works (Jia et al., 2023; Fan et al., 2024). Following, we detail Bad Teacher (Chundawat et al., 2023a) and SCRUB (Kurmanji et al., 2023) results as reference. Finally, in the second part, we report methods that can operate with Support Samples only, including MetaUnlearn.

On CelebA-HQ/20 (table 1), only MetaUnlearn and PGU achieve a higher ToW score than the pretrain model. When unlearning 20 identities, they score 95.7 and 95.6 against 95.5 of the pretrain model, showing the best unlearning performance from the accuracy perspective against tested methods. However, no method, except for MetaUnlearn, substantially improves over the pretrain model on CelebA-HQ/50 (table 1), which scores a ToW of 95.9 vs. 94.8 of pretrain. Interestingly, SSD decreases the forgetting mAP to lower values (82.0 and 70.9) compared to MetaUnlearn (82.6 and 82.5) and PGU (82.8 and 83.7) in both settings, suggesting better forgetting. Yet, it uniformly degrades retain and test sets performance, achieving a lower ToW score (94.3 and 73.7) than MetaUnlearn and PGU. In both CelebA-HQ experiments, JiT performance degradation makes it inapplicable as test mAPs drop by respectively 5.7 and 16.4. By looking at the robustness to membership inference attacks, SSD always achieves the best protection, with MetaUnlearn achieving comparable results (e.g., 0.03% vs. 0.04% in CelebA-HQ/20). Instead, JiT and PGU, show worse results, offering a weaker protection to MIAs (i.e., 0.97% and 0.17% in CelebA-HQ/20). By, factoring in the better accuracy alignment with the retrained model, MetaUnlearn achieves the best performance overall.

Compared to the above case, JiT achieves the best alignment with the retrained model on CelebA/5 (table 2) in accuracy terms (scoring 97.6) while being slightly worse than the best result (MetaUnlearn and PGU) in

Table 1: **Unlearning on CelebA-HQ dataset**. The first two columns report the method name and whether it uses retain and forget sets. Following, we show the average mAP on the retain, forget, and test set, while we show the ToW metric and MIA in the last two columns.

| Method | Access to dataset | mAP | | | ToW ↑ | MIA ↓ |
| --- | --- | --- | --- | --- | --- | --- |
| | | $\mathcal{D}_r$ | $\mathcal{D}_f$ | $\mathcal{D}_{te}$ | | |
| 20 IDENTITIES | | | | | | |
| Pretrain | - | 84.8±0.1 | 83.0±2.9 | 80.7±0.0 | 95.5±0.6 | 0.97% |
| Retrain | - | 84.7±0.2 | 78.6±3.5 | 80.8±0.1 | - | - |
| Bad Teacher (Chundawat et al., 2023a) | ✓ | 84.4±0.1 | 82.3±2.9 | 80.3±0.2 | 95.6±0.9 | 0.04% |
| SCRUB (Kurmanji et al., 2023) | ✓ | 88.1±0.1 | 80.2±2.9 | 81.2±0.2 | 94.7±1.7 | 0.02% |
| JiT (Foster et al., 2024a) | ✗ | 78.2±1.5 | 75.9±4.4 | 75.0±1.2 | 85.7±3.4 | 0.97% |
| PGU (Hoang et al., 2024) | ✗ | 84.6±0.0 | 82.8±2.9 | 80.6±0.1 | 95.6±0.6 | 0.17% |
| SSD (Foster et al., 2024b) | ✗ | 83.4±0.6 | 82.0±3.1 | 79.6±0.5 | 94.3±1.3 | **0.03**% |
| MetaUnlearn | ✗ | 84.5±0.2 | 82.6±2.5 | 80.7±0.1 | **95.7**±0.8 | 0.04% |
| 50 IDENTITIES | | | | | | |
| Pretrain | - | 84.9±0.1 | 83.8±1.2 | 80.8±0.1 | 94.8±0.4 | 6.10% |
| Retrain | - | 84.6±0.2 | 79.1±1.2 | 80.6±0.1 | - | - |
| Bad Teacher (Chundawat et al., 2023a) | ✓ | 83.6±0.1 | 81.9±0.4 | 79.4±0.1 | 95.1±0.9 | 0.49% |
| SCRUB (Kurmanji et al., 2023) | ✓ | 87.0±0.6 | 81.4±0.7 | 81.2±0.3 | 94.7±1.0 | 0.02% |
| JiT (Foster et al., 2024a) | ✗ | 66.3±1.2 | 65.3±0.9 | 64.7±1.3 | 59.2±2.4 | 0.41% |
| PGU (Hoang et al., 2024) | ✗ | 84.6±0.3 | 83.7±1.6 | 80.5±0.3 | 94.9±0.6 | 0.41% |
| SSD (Foster et al., 2024b) | ✗ | 72.7±5.0 | 70.9±5.7 | 70.9±4.5 | 73.7±11.3 | **0.01**% |
| MetaUnlearn | ✗ | 84.1±0.4 | 82.5±1.9 | 80.6±0.3 | **95.9**±0.6 | 0.04% |

Table 2: **Unlearning on CelebA dataset**. The first two columns report the method name and whether it uses retain and forget sets. Following, we show the average mAP on the retain, forget, and test set, while we show the ToW metric and MIA in the last two columns.

| Method | Access to dataset | mAP | | | ToW ↑ | MIA ↓ |
| --- | --- | --- | --- | --- | --- | --- |
| | | $\mathcal{D}_r$ | $\mathcal{D}_f$ | $\mathcal{D}_{te}$ | | |
| 5 IDENTITIES | | | | | | |
| Pretrain | - | 84.4±0.0 | 81.8±2.2 | 80.9±0.1 | 97.2±1.0 | 13.25% |
| Retrain | - | 84.5±0.0 | 79.1±3.2 | 80.9±0.1 | - | - |
| Bad Teacher (Chundawat et al., 2023a) | ✓ | 84.3±0.0 | 79.5±3.4 | 80.7±0.1 | 99.1±0.5 | 22.89% |
| SCRUB (Kurmanji et al., 2023) | ✓ | 87.9±0.1 | 77.5±3.4 | 80.6±0.2 | 94.8±0.7 | 3.61% |
| JiT (Foster et al., 2024a) | ✗ | 84.2±0.1 | 81.1±2.2 | 80.7±0.2 | **97.6**±1.2 | 8.43% |
| PGU (Hoang et al., 2024) | ✗ | 84.4±0.0 | 82.1±2.1 | 80.9±0.1 | 96.9±1.5 | **7.23**% |
| SSD (Foster et al., 2024b) | ✗ | 23.1±0.2 | 30.9±2.0 | 23.1±0.1 | 8.4±0.9 | 13.25% |
| MetaUnlearn | ✗ | 84.4±0.0 | 81.9±2.2 | 80.9±0.1 | 97.2±1.0 | 9.64% |
| 10 IDENTITIES | | | | | | |
| Pretrain | - | 84.5±0.1 | 83.0±4.1 | 80.9±0.1 | 97.3±0.5 | 29.76% |
| Retrain | - | 84.4±0.1 | 80.4±4.5 | 80.9±0.1 | - | - |
| Bad Teacher (Chundawat et al., 2023a) | ✓ | 84.2±0.1 | 82.2±3.7 | 80.5±0.1 | 97.6±1.0 | 33.93% |
| SCRUB (Kurmanji et al., 2023) | ✓ | 87.9±0.1 | 79.9±4.7 | 80.6±0.1 | 95.8±0.3 | 1.20% |
| JiT (Foster et al., 2024a) | ✗ | 83.9±0.4 | 82.5±4.4 | 80.4±0.3 | 96.9±0.9 | 23.21% |
| PGU (Hoang et al., 2024) | ✗ | 84.5±0.0 | 82.9±4.3 | 80.9±0.1 | **97.4**±0.3 | 25.00% |
| SSD (Foster et al., 2024b) | ✗ | 26.9±4.3 | 30.1±5.2 | 26.9±4.2 | 10.0±3.2 | **10.12**% |
| MetaUnlearn | ✗ | 84.4±0.0 | 82.9±4.3 | 80.9±0.2 | **97.4**±0.3 | 19.05% |

CelebA/10 (table 2). SSD performance degradation (test mAP of 23.1 and 26.9) makes it inapplicable in real-world scenarios, although it achieves the best MIA in CelebA/10. MetaUnlearn and PGU perform similarly

Table 3: **Unlearning on MUFAC dataset**. The first two columns report the method name and whether it uses retain and forget sets. Following, we show the average Accuracy on the retain, forget, and test set, while we show the ToW metric and MIA in the last two columns.

| Method | Access to dataset | Accuracy | | | ToW ↑ | MIA ↓ |
|---|---|---|---|---|---|---|
| | | $\mathcal{D}_r$ | $\mathcal{D}_f$ | $\mathcal{D}_{te}$ | | |
| 5 IDENTITIES | | | | | | |
| Pretrain | - | 99.8±0.1 | 99.3±1.0 | 78.2±0.7 | 70.3±10.3 | 0.04% |
| Retrain | - | 99.7±0.1 | 70.0±11.4 | 78.1±0.4 | - | |
| Bad Teacher (Chundawat et al., 2023a) | ✓ | 99.7±0.0 | 67.3±8.3 | 77.5±1.1 | 94.4±4.4 | 0.00% |
| SCRUB (Kurmanji et al., 2023) | ✓ | 100.0±0.0 | 89.0±3.8 | 78.0±0.8 | 80.1±9.9 | 0.02% |
| JiT (Foster et al., 2024a) | ✗ | 99.7±0.1 | 99.3±1.0 | 77.8±0.8 | 70.1±10.3 | 0.02% |
| PGU (Hoang et al., 2024) | ✗ | 99.8±0.1 | 99.3±1.0 | 78.1±0.6 | 70.3±10.3 | 0.04% |
| SSD (Foster et al., 2024b) | ✗ | 15.2±1.6 | 14.4±4.0 | 15.0±1.1 | 2.5±0.5 | **0.01%** |
| MetaUnlearn | ✗ | 99.6±0.3 | 98.5±1.0 | 78.0±0.9 | **70.8±10.3** | 0.03% |
| 10 IDENTITIES | | | | | | |
| Pretrain | - | 99.9±0.0 | 99.7±0.5 | 77.4±0.7 | 76.7±7.6 | 0.03% |
| Retrain | - | 99.6±0.4 | 76.8±7.4 | 77.4±0.7 | - | |
| Bad Teacher (Chundawat et al., 2023a) | ✓ | 99.7±0.0 | 67.9±8.0 | 77.2±0.1 | 90.2±0.7 | 0.00% |
| SCRUB (Kurmanji et al., 2023) | ✓ | 99.8±0.0 | 95.7±2.7 | 78.3±0.3 | 80.2±5.0 | 0.02% |
| JiT (Foster et al., 2024a) | ✗ | 99.7±0.1 | 99.7±0.5 | 76.9±0.9 | 76.4±7.3 | 0.03% |
| PGU (Hoang et al., 2024) | ✗ | 99.5±0.3 | 96.8±2.3 | 76.9±1.1 | 79.0±9.1 | 0.04% |
| SSD (Foster et al., 2024b) | ✗ | 12.3±4.3 | 15.3±2.4 | 12.5±3.5 | 1.7±0.6 | **0.01%** |
| MetaUnlearn | ✗ | 96.3±3.5 | 91.8±7.1 | 74.1±3.2 | **79.1±6.4** | 0.04% |

in both CelebA tests, scoring the best ToW in CelebA/10 (97.4). However, MetaUnlearn slightly outperforms PGU in CelebA/5 (97.2 vs. 96.9), achieving better alignment with the retrained model. By investigating the MIA, PGU and JiT achieve the best and second best results (7.23% and 8.43%) on CelebA/5, while MetaUnlearn is slightly outperformed (9.64%). Yet, MetaUnlearn achieves the best MIA on CelebA/10 among methods that preserve a sufficiently high mAP, scoring 19.05% vs. 23.21% of JiT (the second best). MetaUnlearn is overall more consistent in CelebA experiments than existing baselines. It shows the same average ToW with JiT (97.3 vs. 97.25) across the two experiments but a lower average MIA of 14.3% against 15.8% of PGU.

Finally, the MUFAC dataset demonstrates the most challenging of the three (table 3). We believe its difficulty is caused by the large age gap between different pictures of the same identity, especially for young people. All tested methods struggle to remove target identities effectively from the ToW perspective. SSD cannot identify important parameters for the forget set, scoring a ToW of 2.5 and 1.7 in MUFAC/5 and MUFAC/10. Instead, JiT always fails to improve the ToW over the pretrain model, suggesting little-to-no unlearning. On the contrary, PGU and MetaUnlearn both improve over the pretrain model on MUFAC/10 by 2.3 and 2.4 points respectively. However, only MetaUnlearn manages to improve over the original model scoring a ToW of 70.8. Additionally, among methods that preserve model utility (JiT, PGU, MetaUnlearn), only MetaUnlearn shows a reduction in the forget set accuracy (e.g., -0.8% on MUFAC/5, - 7.9% on MUFAC/10), further suggesting that identities are at least partly unlearned. Regardless, the attack success rate (MIA) is very low for all methods, where values are very close to random guessing (0.01%), which suggests good privacy protection from membership inference attacks.

Although MetaUnlearn struggles to outperform existing baselines in CelebA, it is more consistent than tested approaches, showing the best or comparable performance in all six configurations. The next section (section 5.3), compares MetaUnlearn performance against other methods when unlearning extremely small forget set sets, i.e. when the number of unlearning identities equals one.

Table 4: **Unlearning one identity on CelebA-HQ and CelebA dataset**. The first two columns report the method name and whether it uses retain and forget sets. Following, we show the average mAP on the retain, forget, and test set, while we show the ToW metric and MIA in the last two columns.

| Method | Access to dataset | mAP | | | ToW ↑ | MIA ↓ |
|---|---|---|---|---|---|---|
| | | $\mathcal{D}_r$ | $\mathcal{D}_f$ | $\mathcal{D}_{te}$ | | |
| CELEBA-HQ | | | | | | |
| Pretrain | - | $84.5{\pm}0.3$ | $90.9{\pm}8.9$ | $80.7{\pm}0.1$ | $95.8{\pm}2.7$ | 0.01% |
| Retrain | - | $84.6{\pm}0.2$ | $87.2{\pm}9.9$ | $80.7{\pm}0.1$ | - | - |
| Bad Teacher (Chundawat et al., 2023a) | ✓ | $84.5{\pm}0.3$ | $89.9{\pm}10.3$ | $80.7{\pm}0.1$ | $96.8{\pm}2.9$ | 0.03% |
| SCRUB (Kurmanji et al., 2023) | ✓ | $88.0{\pm}0.2$ | $86.8{\pm}8.9$ | $81.1{\pm}0.2$ | $95.1{\pm}0.4$ | 0.01% |
| JiT (Foster et al., 2024a) | ✗ | $84.4{\pm}0.2$ | $91.1{\pm}8.7$ | $80.5{\pm}0.2$ | $95.6{\pm}2.8$ | 0.14% |
| PGU (Hoang et al., 2024) | ✗ | $84.4{\pm}0.3$ | $89.9{\pm}8.5$ | $80.6{\pm}0.0$ | $\underline{96.8}{\pm}1.8$ | **0.01**% |
| SSD (Foster et al., 2024b) | ✗ | $25.2{\pm}0.5$ | $70.8{\pm}17.1$ | $25.6{\pm}0.4$ | $15.2{\pm}1.9$ | 0.05% |
| MetaUnlearn | ✗ | $83.7{\pm}0.0$ | $88.2{\pm}9.1$ | $80.1{\pm}0.3$ | **97.4**${\pm}0.8$ | **0.01**% |
| CELEBA | | | | | | |
| Pretrain | - | $84.5{\pm}0.1$ | $83.5{\pm}7.1$ | $80.9{\pm}0.1$ | $96.6{\pm}1.5$ | 5.26% |
| Retrain | - | $84.5{\pm}0.1$ | $80.2{\pm}6.7$ | $80.9{\pm}0.1$ | - | - |
| Bad Teacher (Chundawat et al., 2023a) | ✓ | $84.4{\pm}0.1$ | $83.7{\pm}6.2$ | $80.9{\pm}0.1$ | $96.5{\pm}1.0$ | 0.32% |
| SCRUB (Kurmanji et al., 2023) | ✓ | $88.0{\pm}0.0$ | $80.4{\pm}5.5$ | $80.6{\pm}0.1$ | $95.2{\pm}0.6$ | 0.01% |
| JiT (Foster et al., 2024a) | ✗ | $84.5{\pm}0.1$ | $82.5{\pm}6.9$ | $80.9{\pm}0.1$ | **97.6**${\pm}0.4$ | 5.26% |
| PGU (Hoang et al., 2024) | ✗ | $84.4{\pm}0.1$ | $83.4{\pm}7.0$ | $80.9{\pm}0.1$ | $\underline{96.8}{\pm}1.2$ | 0.26% |
| SSD (Foster et al., 2024b) | ✗ | $22.7{\pm}0.2$ | $56.1{\pm}4.8$ | $22.7{\pm}0.3$ | $12.1{\pm}1.7$ | **0.01**% |
| MetaUnlearn | ✗ | $84.5{\pm}0.1$ | $83.5{\pm}7.2$ | $80.9{\pm}0.1$ | $96.7{\pm}1.5$ | **0.01**% |

## 5.3 Evaluation on extremely small forget set

Tables 1 and 2 compare MetaUnlearn and baselines in the proposed 1-SHUI task by varying the number of unlearning identities. In all experiments, we always considered the number of forgetting identities to be greater than one. This section tests the robustness of the approaches when only one user asks to be forgotten, therefore, experimenting in the case where $N_{\mathcal{S}} = 1$. Results are computed following the same evaluation procedure as tables 1 and 2 and are reported in table 4. Our method achieves the best ToW score (97.4) in CelebA-HQ, showing great alignment with the retrained model. Yet, it performs close to the second-best PGU (Hoang et al., 2024) (96.7 vs 96.8) in CelebA. Nonetheless, it strongly preserves privacy by scoring the lowest MIA rate on both datasets. JiT (Foster et al., 2024a) scores the best ToW in CelebA (97.6), while achieving the same MIA as the original model or worse, failing to unlearn the sensitive information. PGU, instead, achieves worse or comparable results on both datasets from both metrics perspectives. Finally, SSD strongly degrades model performance after unlearning, making it unusable. Overall, our method achieves the best ToW or is comparable with existing baselines while always achieving the best MIA, therefore better preserving the privacy of the sensitive data.

## 5.4 When is unlearning the hardest?

In evaluating MetaUnlearn and existing methods in 1-SHUI, we noticed how some identities were easier to unlearn than others. We conjecture that this behavior relates to the dissimilarity between Support Samples and forget set images of the same identity (see figure 6), i.e. unlearning an identity that shows a large gap between the Support Samples and images in the forget set might be harder. To quantify this, we computed the accuracy difference between the unlearned and retrained models for each unlearning identity, averaging the results of the top-3 unlearning methods on three seeds. We then compute the Euclidean distance in the feature space between each Support Sample and the centroid of the forget data of its corresponding identity. Finally, we produced a bar plot, grouping identities with similar forget data-support sample distance and sorting the distance from left to right in increasing order.

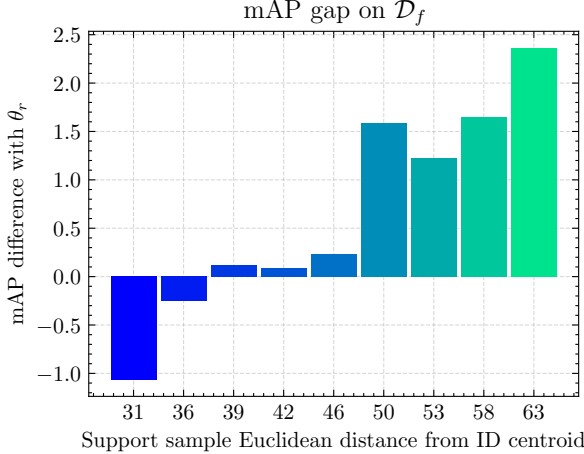

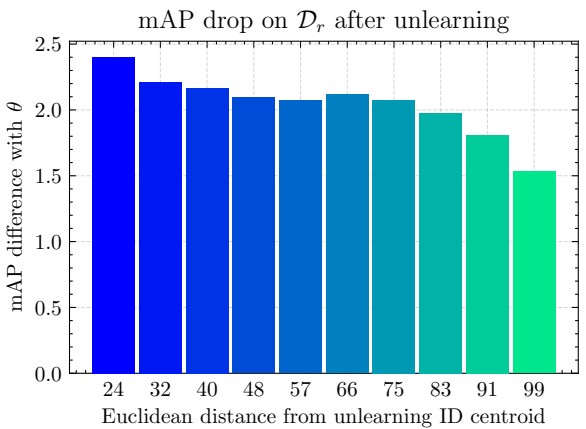

Figure 4: **Unlearning hardness vs. Support Sample distance.** As the Support Sample distance from the identity centroid increases, the accuracy gap with the retrained model grows.

Figure 5: **Performance drop vs. distance from forget set.** As retain identities get closer to forget identities, retain samples mAP drops compared to the pretrain model.

Figure 4 shows the results of our analysis. The unlearned model accuracy on forget data struggles to achieve retrained model values as the distance increases, going from a difference of -1.1 to 2.4 in mAP terms. We also outline that if the Support Samples are close enough to identity centroids, the forget accuracy can be lower than the retrain model value, suggesting a possible *Streisand Effect* (Chundawat et al., 2023a; Foster et al., 2024b). The Streisand Effect occurs when unlearning is too aggressive and the accuracy of the forget set drops way below the retrained model one. When the model confidence on unlearned samples is too low compared to unseen samples, it can suggest to the attacker that the forget set was previously part of the training set and it was unlearned, leaking membership information.

## 5.5 When does unlearning harm performance?

Analogously to the previous case (section 5.4), we noticed that, in both datasets, most methods demonstrated a drop in performance from the original model on the retain data, especially on CelebA-HQ. We hypothesize that the closer a retain identity is to a forget one, the higher the chance of an accuracy drop on retain data compared to the original model. Thus, we analyzed whether the distance between forget identities' centroids and retain ones correlates with a performance drop. We computed the mAP difference from the pretrain model for each retain identity, averaging three seeds and using the top-3 methods on CelebA-HQ/50. We binned accuracy drops based on identity distances, as in section 5.4. Figure 4 shows the result of our experiment. The closer a retain sample is to a forget sample, the more the mAP drops from the original model, which ranges from 2.4 to 1.5 as the distance grows. Consequently, preserving the original performance becomes easier as the distance between forget and retain samples increases.

## 5.6 Ablation studies

This section shows a comprehensive ablation study of MetaUnlearn auxiliary loss terms, inputs to the meta-loss, and simulated requests size. We show the performance in ToW terms, due to its much cheaper computational cost than MIA while achieving correlated results (Zhao et al., 2024b).

**Auxiliary loss function.** Table 5 shows the ablation study of the auxiliary loss used to train MetaUnlearn. For completeness, we report the performances of our approach in all four configurations, averaging over three different seeds. In the first row, we show ToW scores when naïvely applying SCRUB (Kurmanji et al., 2023) loss function to measure the unlearning effectiveness. As a result, either the approach performs sub-optimally or fails to achieve meaningful ToW scores. We hypothesize that, as SCRUB loss is numerically unstable if

Table 5: **Ablating MetaUnlearn auxiliary loss.** The first column reports the loss function used, i.e.SCRUB loss or different combinations of equation 5. Then, we report ToW scores for settings we describe in section 5.1. Note that OOM is short for "out-of-memory".

| Loss Type | ToW ↑ | | | |
|---|---|---|---|---|
| | CelebA-HQ/20 | CelebA-HQ/50 | CelebA/5 | CelebA/10 |
| SCRUB (Kurmanji et al., 2023) | 94.3±0.2 | OOM | 51.7±31.4 | 55.4±15.0 |
| First term | 8.3±0.7 | 32.4±33.4 | 7.6±0.8 | 6.9±0.6 |
| First term + Accuracy | 8.1±0.7 | 8.2±0.2 | 7.6±0.8 | 6.9±0.7 |
| Second term | 95.7±0.9 | 95.1±0.3 | 97.1±1.1 | **97.4**±0.4 |
| Second term + Accuracy | 95.7±1.1 | 95.7±1.0 | **97.2**±1.0 | 97.3±0.3 |
| First & Second terms | 95.6±1.0 | 95.4±0.3 | 97.1±1.0 | 97.3±0.4 |
| First & Second terms + Accuracy | **95.8**±0.7 | **95.9**±0.6 | **97.2**±1.0 | **97.4**±0.3 |

Table 6: **Ablating the inputs to MetaUnlearn.** The first column lists all possible inputs to the meta-loss, while in the remaining four we report the ToW metric on the described settings.

| Input | | | | ToW ↑ | | | |
|---|---|---|---|---|---|---|---|
| Logits | Features | IDs | Targets | CelebaA-HQ/20 | CelebaA-HQ/50 | CelebA/5 | CelebaA/10 |
| ✓ | | | | 95.0±1.4 | 93.0±3.0 | **97.3**±1.2 | 97.3±0.4 |
| ✓ | ✓ | | | **96.1**±0.9 | 95.4±0.8 | 97.2±1.1 | 97.3±0.3 |
| ✓ | ✓ | ✓ | | 95.7±0.7 | 95.8±0.5 | 97.2±1.1 | 97.2±0.3 |
| ✓ | ✓ | ✓ | ✓ | 95.7±0.8 | **95.9**±0.6 | 97.2±1.0 | **97.4**±0.3 |

not optimized in an alternate way (Kurmanji et al., 2023; Chen & Yang, 2023), using it to measure the meta-loss error can similarly be problematic. In one case (CelebA-HQ/50) we experienced out-of-memory issues as SCRUB requires the model to forward retain and forget images through the original and unlearned model, largely increasing the memory footprint in a second-order optimization such as ours.

In the second and third rows, we show MetaUnlearn performance when using only the first term of equation 5, where TOW scores are close to zero. The forget error gets progressively closer to the validation error when the second term is omitted. However, the validation empirical error is not bounded, which causes performance degradation (e.g., ToW of 6.9 in CelebA/10). Instead, using the second term of the loss offers more stability as it aligns the forget error with the loss on the validation set obtained using the original model, whose weights are frozen (97.4 of ToW in CelebA/10). When scaling losses by their mAP values (see section 4.1), using both terms provides consistently better results among all four settings, always achieving the best performance.

**Input to the meta-loss.** Table 6 reports the results of an ablation study about the inputs provided to the meta-loss function. The first row shows results by forwarding only output logits to the meta-loss. MetaUnlearn struggles in CelebA-HQ dataset, achieving a ToW score of 95.0 and 93.0 (against 95.5 and 94.8 of the pretrain model), while we find it sufficiently good in CelebA. As the second row highlights, by concatenating model features to logits, CelebA-HQ performances improve to 96.1 and 95.4, surpassing the pretrain model. When also concatenating identities and ground truth annotations, the ToW slightly grows on average. Therefore, in our experiments, we use all four inputs to MetaUnlearn.

Section 4.1 describes how we simulate multiple unlearning requests using a fixed size $N_{\mathcal{S}}$, which corresponds to the one used during evaluation (i.e. we set the simulated unlearning size to 20 for the CelebA-HQ/20 experiment). This choice follows practical considerations: e.g. if the unlearning set size is larger, one can split it to match the desired size while, if it is smaller, one can queue the requests until the desired number is reached. While this choice is sound in scenarios where the user can manage the unlearning set size, there might be cases where the size of real unlearning requests is unknown a priori. In this section, we test whether MetaUnlearn is affected by shifts between differences in number of requests between training and test. To

assess this, we test our method when the size of the simulated and real unlearning requests differ. We perform four experiments: on CelebA-HQ 20 using $N_S = 50$ during training, and vice versa (i.e., $N_S = 20$ for CelebA-HQ 50). We also test it on CelebA-10 with $N_S = 5$ during training and vice versa (i.e., $N_S = 10$ for CelebA-5). Table 7 summarizes the results, where *Reference* refers to the number of simulated requests matching those seen at evaluation time. In general, MetaUnlearn shows a very low discrepancy with the reference, i.e. with differences between 0.1 (CelebA-HQ 20 identities, CelebA 10 identities) and 0.3 (CelebA 5 identities) ToW. Although the best results are always achieved when the size of the simulated requests during training matches those of the test, MetaUnlearn achieves comparable performance also in case of discrepancies, showing its robustness to the choice of $N_S$.

## 6    Conclusions

This work is the first to investigate the unlearning of randomized samples in the data absence scenario. We present a novel task we call One-Shot Unlearning of Personal Identities (1-SHUI) to evaluate algorithms that perform unlearning when the original dataset is inaccessible. 1-SHUI focuses on identity unlearning, a specific case of random unlearning where methods must forget all data associated with the unlearning identity. To circumvent the lack of training data, we assume that users asking to be forgotten, provide one of their pictures (Support Sample) to aid unlearning. To tackle this challenging task, we propose a learning-to-unlearn approach (MetaUnlearn) that learns to forget identities by relying on the provided Support Sample. MetaUnlearn simulates multiple unlearning requests to optimize a parameterized loss function while data is still available. This results in a loss function that can forget identities without relying on the original training data. Experiments on three datasets show that MetaUnlearn achieves good forgetting results on 1-SHUI, being a first step toward tackling this challenging task.

Table 7: **Robustness to different unlearning set sizes**. Even when the simulated and real unlearning set sizes do not match, MetaUnlearn performance is comparable to the best case where they match.

| CelebA-HQ | | CelebA | |
|---|---|---|---|
| Training | ToW ↑ | Training | ToW ↑ |
| 20 IDENTITIES | | 5 IDENTITIES | |
| Reference | 95.7±0.8 | Reference | 97.2±1.0 |
| 50 Identities | 95.6±1.0 | 10 Identities | 96.9±1.0 |
| 50 IDENTITIES | | 10 IDENTITIES | |
| Reference | 95.9±0.6 | Reference | 97.4±0.3 |
| 20 Identities | 95.7±0.6 | 5 Identities | 97.3±0.3 |

**Limitations and future works.** One limitation of our work is that we focus on a single unlearning request. As confirmed by our experiments, 1-SHUI is extremely challenging, and accounting for multiple unlearning requests could have made the evaluation tough. Therefore, exploring unlearning algorithms that account for multiple unlearning requests in the data absence scenario is a natural extension of this work.

Due to their wide availability and fine-grained attribute annotation, we constrained our analysis to face datasets. At the same time, it would be interesting to test MetaUnlearn and baselines in more challenging settings, such as images of the full body or even images annotated with multiple identities. Moreover, 1-SHUI could be transferred to cases beyond identities, such as specific objects and/or animals. While these are less interesting use cases from a regulation point of view, they might be considered in future works.

**Acknowledgements.** We acknowledge the CINECA award under the ISCRA initiative for the availability of high-performance computing resources and support. Elisa Ricci and Massimiliano Mancini are supported by the MUR PNRR project FAIR - Future AI Research (PE00000013), funded by NextGeneration EU. Elisa Ricci is also supported by the EU projects AI4TRUST (No.101070190) and ELIAS (No.01120237) and the PRIN project LEGO-AI (Prot.2020TA3K9N). Thomas De Min is funded by NextGeneration EU. This work has been supported by the French National Research Agency (ANR) with the ANR-20-CE23-0027.

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

# A   Implementation details

For all experiments, we used a ViT-B/16 (Dosovitskiy et al., 2021) pre-trained on ImageNet (Russakovsky et al., 2015). We fine-tuned ViT on CelebA and CelebA-HQ for 30 epochs, using SGD with a learning rate of $1 \times 10^{-3}$ and momentum 0.9. The learning rate is initially warm-upped for the first two training epochs and decayed following a cosine annealing schedule for the rest of the optimization. We regularize the optimization using weight decay with penalties of $1 \times 10^{-3}$ and $1 \times 10^{-4}$ for CelebA-HQ and CelebA. Additionally, we augment input images with RandomResizedCrop and RandomHorizontalFlip to further regularize (He et al., 2016). The same optimization configuration is used for both model pretraining and retraining. We used mixed precision for all experiments using the "brain floating point 16" to speed up training. Furthermore, a single A100 Nvidia GPU was used for all experiments.

The meta-loss was trained for 3 epochs, except for one experiment in MUFAC, where each epoch consists of iteratively unlearning all identities in batches of $N_{\mathcal{S}}$ IDs. Therefore, at each iteration, we sampled without replacement $N_{\mathcal{S}}$ IDs, unlearned those identities using $h_\phi$, computed the auxiliary loss function $\mathcal{A}$ (equation 6), and backpropagated the error back to $h_\phi$. Then, we discarded the unlearned network and repeated the above operations until all identities were unlearned, and cycle for three epochs. We used Adam optimizer (Kingma, 2014) with AMSgrad (Reddi et al., 2018) and no weight decay. The learning rate ($\alpha$) was choosed from $\{10^{-4}, 10^{-3}, 10^{-2}\}$ and was decayed following a cosine annealing schedule. Instead, the meta-learning rate ($\eta$), used when computing the unlearning step, was chosen from $\{10^{-3}, 0.1\}$. The meta-loss architecture was composed of a linear encoder that mixes input features and projects them from the input size $|\mathcal{Y}| + D + 64 + |\mathcal{Y}|$ to the hidden size (512), where $|\mathcal{Y}|$ is the cardinality of the output space, $D$ is the backbone output size, and 64 is the chosen size of the identity embedding table. Then, we subsequently applied a LayerNorm (Ba, 2016) layer, a linear projection, and a non-linear activation function (Hendrycks & Gimpel, 2016). We used Dropout (Srivastava et al., 2014) for regularization with a probability of 0.5 and another LayerNorm before computing the output projection from the hidden size to a scalar value. Finally, as the loss function must output positive values only, we applied a softplus transformation. Tables 8a and 8b summarize the training details of MetaUnlearn and the architecture used.

**On the absence of $\mathcal{D}_r$ in $\mathcal{A}$.**   We empirically noticed that directly enforcing performance alignment with $\mathcal{D}_r$ hinders unlearning. We argue that since the size of the retain set is usually much bigger than the forget set (Foster et al., 2024b), $N_r \gg N_f$, at each simulation $\mathcal{D}_r$ differs from $\mathcal{D}_{tr}$ by a small number of samples. In other words, the probability that one sample is chosen for the unlearning request is close to zero. Consequently, while forcing an alignment on $\mathcal{D}_r$, $h_\phi$ finds the alignment with $\mathcal{D}_{tr}$ as a shortcut, hindering unlearning. Empirically, we noticed that adding an alignment constraint on $\mathcal{D}_r$ resulted in an unlearned network that performed closely to the original one, hardly showing signs of unlearning.

Table 8: **MetaUnlearn training details and architecture.** Table 8a report the value assigned tol each hyperparameter. Table 8b instead, presents MetaUnlearn architecture.

<div>

(a) **Training details**.

| Component | Value |
|---|---|
| input_size | $|\mathcal{Y}| + D + 64 + |\mathcal{Y}|$ |
| hidden_size | 512 |
| identity_embed_dim | 64 |
| dropout_prob | 0.5 |
| regression_loss | smooth_l1_loss |
| meta_lr ($\eta$) | $\{10^{-3}, 0.1\}$ |
| optimizer | Adam |
| lr ($\alpha$) | $\{10^{-4}, 10^{-3}, 10^{-2}\}$ |
| epochs | $\{3, 10\}$ |
| scheduler | cosine |
| amsgrad | True |
| amp | bfloat16 |

(b) **Meta-loss architecture**.

| Layer | Component |
|---|---|
| 0 | Linear(input_size, hidden_size) |
| 1 | LayerNorm(hidden_size) |
| 2 | Linear(hidden_size, hidden_size) |
| 3 | GeLU() |
| 4 | Dropout(0.5) |
| 5 | LayerNorm(hidden_size) |
| 6 | Linear(hidden_size, 1) |
| 7 | Softplus() |

</div>

# B    Membership Inference Attack

To evaluate the privacy protection level of unlearning methods, we rely on the recently proposed RMIA (Zarifzadeh et al., 2024). Contrary to previous membership inference attacks (Carlini et al., 2022; Shokri et al., 2017) that are less efficient and require different shadow models to achieve satisfactory results, RMIA performs attacks with just one shadow model in its cheapest formulation. To test the unlearning capabilities of each approach, we retrained the original model using only the retain data. Then, we ran each unlearning algorithm on the retrained model to account for parameter changes introduced by specific algorithms. We compute the membership score as detailed by Zarifzadeh et al. (2024):

$$\mathrm{LR}_{\theta_u}(x, z) = \left( \frac{\mathrm{Pr}(x \mid \theta_u)}{\mathrm{Pr}(x \mid \theta'_u)} \right) \cdot \left( \frac{\mathrm{Pr}(z \mid \theta_u)}{\mathrm{Pr}(z \mid \theta'_u)} \right)^{-1}, \tag{8}$$

$$\mathrm{Score}_{\mathrm{MIA}}(x; \theta_u) = \Pr_{z \sim \mathcal{D}} \left( \mathrm{LR}_{\theta_u}(x, z) \geq \gamma \right), \tag{9}$$

where $x$ is a target sample we want to infer membership on, $\theta'_u$ is the model retrained and unlearned (reference model), $z$ is a random sample drawn from the dataset, and $\gamma$ is a hyperparameter. Following Zarifzadeh et al. (2024), we predict membership as:

$$\mathrm{MIA}(x; \theta_u) = \mathbb{1}\left\{ \mathrm{Score}_{\mathrm{MIA}}(x; \theta_u) \geq \beta \right\}. \tag{10}$$

We set $\beta$ such that the FPR = 0.01% (False Positive Rate), and we report the corresponding TPR (True Positive Rate) in the results. We refer to the original paper (Zarifzadeh et al., 2024) for further details.

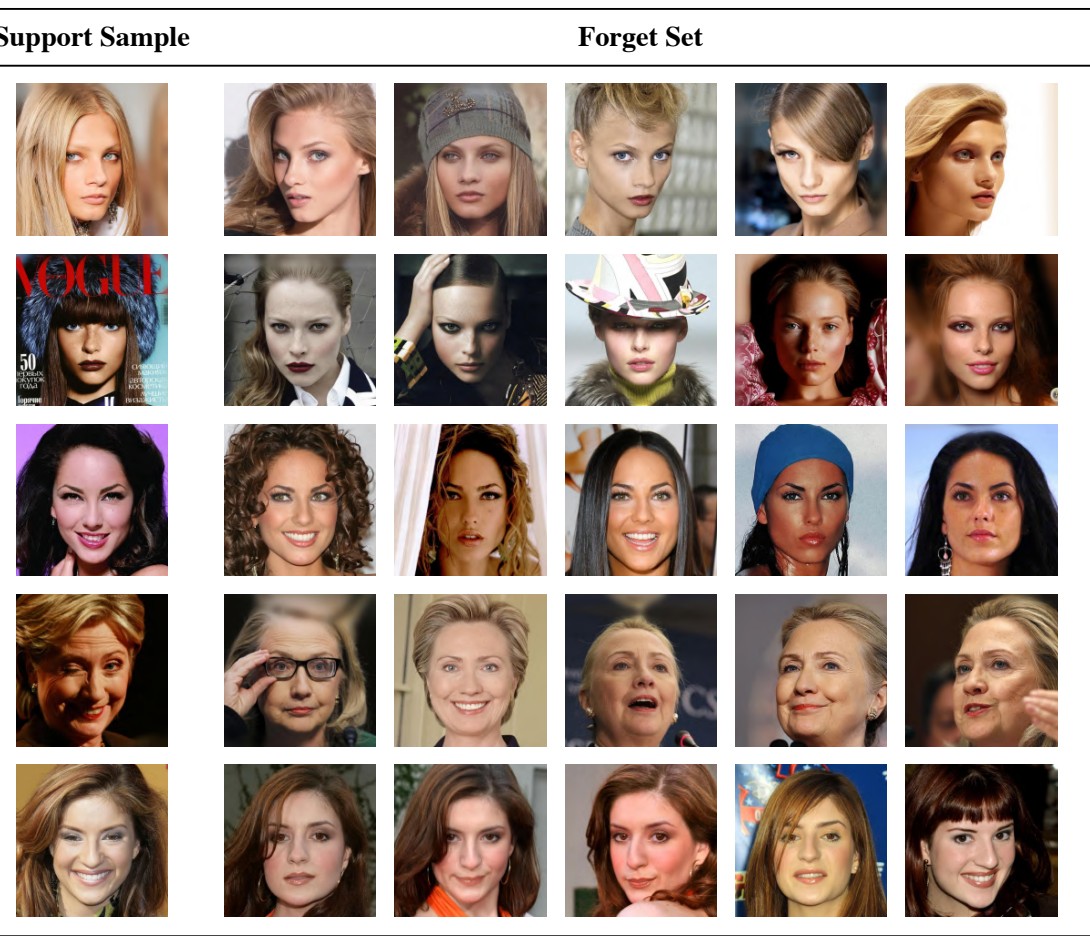

Figure 6: **Visualizing unlearning hardness**. This Figure portrays the five identities with the highest embedding distance between the Support Sample and the forget set. Images are taken from CelebA-HQ.

