# OpenReview forum: "Unlearning Personal Data from a Single Image"
_TMLR — Accepted by TMLR_

### Review · Reviewer_Hsay · 2024-12-17

**Summary Of Contributions:**

This paper emphasizes the problem of unavailable realistic training data which is often required in existing unlearning evaluation benchmarks.
1-SHUI benchmark is proposed to evaluate machine unlearning without requiring the training data.
Existing MU methods struggle to tackle the 1-SHUI task, while this paper designs the MetaUnlearn method to tackle the problem, showing its effectiveness on the new benchmark 1-SHUI.

**Audience:**

Yes

**Broader Impact Concerns:**

Besides the facial recognition application, this work will be more impactful if there are any other possible applications of such one-shot  machine unlearning tasks.

**Claims And Evidence:**

Yes

**Requested Changes:**

I think the most important part is the point 2 of the weakness. In the new setting, what data is available for the machine unlearning?

**Strengths And Weaknesses:**

Strengths:

1. This new benchmark is closer to the realistic scenario where the model training data is not available but the model still needs to be evaluated for the machine unlearning.
Moreover, they can control the unlearned information according to the given support image, which seems to be reasonable in reality.

2. The proposed method seems to be effective based on the experiment results. The detailed experiment discussion and analysis are beneficial to understanding the method better.

Weakness:
1. The Related Work section [1,2,3] mentions some methods for unlearning data in scenarios where the original data is not available.
However, these methods are not included in the main discussion of the paper.
Since these methods are important, the authors should mention them in the Introduction and clearly explain how the newly proposed benchmark is different.
This will help readers understand how this work fits into the existing research and what makes it unique.

2. I am a bit confused about the setting. The paper emphasizes that the original training set is not needed for the machine unlearning.
However, in Figure 3 of the method pipeline, the forget sets $D_f$ appear for the loss calculation. In my understanding, the forget setting is part of the training data, so it is also not available in training.
If my understanding is not correct, please add more clarification in the method section to make it clear which part is available in the machine unlearning or not.

3. This method is not specifically designed for face attribute recognition, so I think it is better to include more various datasets to evaluate the method's effectiveness. This can also benefit the benchmark, which can be proven to have more applications other than face recognition.

4. It is better to clarify what is $L_{\rm task}$ in Eq. 5 for better understanding at least in the Appendix.


References:

[1] Zero-Shot Machine Unlearning

[2] Fast Yet Effective Machine Unlearning

[3] Zero-Shot Class Unlearning in CLIP with Synthetic Samples

---

> ### Author Response · Authors · 2025-01-11
>
> We thank the reviewer for his comments and appreciate that he recognized the effectiveness of our approach and its relevance for realistic scenarios. We would like to address his concerns as follows:
> 1. We thank the reviewer for the suggestion. We revised Section 1 of the paper to emphasize that existing approaches are constrained to class-unlearning scenarios due to their reliance on reconstruction-based algorithms. These algorithms cannot reconstruct specific instances and therefore, perform random data unlearning.
> 2. As the reviewer correctly points out, the training set (and, thus, the forget set) is not needed to unlearn identities and it is assumed unavailable at the unlearning time. Our method consists of three steps: (i) we train the original model on the downstream task (e.g., training the model to recognize face attributes or age); (ii) before the training data becomes unavailable and after the model training, we can use the full dataset to optimize the meta-loss function (Section 4.1 of the revised paper), which learns to unlearn identities by only relying on Support Samples, and subsequently discard the data; (iii) when unlearning requests arrive (in the form of Support Samples), we can use the trained loss to unlearn identities without accessing the original dataset. We improved Sections 4.1 and 4.2 of the manuscript highlighting that our proposed loss is learned when data is still available (after model training), while unlearning is performed by accessing the Support Set only. Furthermore, we updated Figure 3 enhancing the three different unlearning stages.
> 3. In the paper, we focused our experiments on face attribute recognition as this task is particularly subject to privacy constraints and relatively understudied in machine unlearning. To further demonstrate the validity of our approach, we performed additional identity unlearning experiments on an age classification model. We used the MUFAC dataset proposed in [1] and followed the same evaluation procedure of CelebA and CelebA-HQ. Table 3 of the revised paper reports the result of our experiment. Overall, our method achieves the best ToW (i.e., 70.8 vs 70.3 in 5 IDs unlearning and 79.1 vs 79.0 in 10 IDs unlearning, when compared to the second-best) in both tested configurations despite the surprisingly challenging dataset. All methods protect against membership inference attacks, scoring low TPR@0.01%FPR. Finally, while we focused on identity unlearning due to the privacy concerns related to personal information, future works may explore similar strategies for other types of unlearning scenarios (e.g., specific animals, objects) as highlighted in Section 6.
> 4. We thank the reviewer for the suggestion. We updated Section 4.1 of the revised paper, highlighting that $L_{task}$ is the loss function used to train the original model on the downstream task.
>
> [1] Choi, Dasol, and Dongbin Na. "Towards machine unlearning benchmarks: Forgetting the personal identities in facial recognition systems." arXiv preprint arXiv:2311.02240 (2023).

---

> > ### Comment · Reviewer_Hsay · 2025-01-21
> > **Reply**
> >
> > Thank you to the authors for providing a detailed explanation, which has addressed most of my concerns.

---

### Review · Reviewer_noQy · 2024-12-24

**Summary Of Contributions:**

This paper proposes a novel task called **1-SHUI (One-Shot Unlearning of Personal Identities)**, focusing on random-sample unlearning (particularly personal identities) under the scenario where the original training data is inaccessible. The key contributions are:
- It introduces a **one-shot unlearning** setup, where each identity to be forgotten is represented by only a single “Support Sample,” forming a new benchmark and evaluation protocol.
- It presents a **meta-learning**-based approach (**MetaUnlearn**), which learns a meta-loss to facilitate forgetting under these extremely limited conditions.
- Experimental results on CelebA and CelebA-HQ show that many existing unlearning methods struggle when the training data is inaccessible, whereas MetaUnlearn achieves a better balance between forgetting and retaining.

Overall, the paper addresses a very practical unlearning scenario and provides an initial feasible solution. This work has implications for privacy and regulatory compliance in machine learning.

---

**Audience:**

Yes

**Broader Impact Concerns:**

None.

**Claims And Evidence:**

Yes

**Requested Changes:**

1. **Add more ablation studies on the meta-loss**
   In response to Weakness (1), conduct a comparison where a standard loss function replaces $\(h_\phi(\cdot)\)$. This will clarify the meta-loss's impact.

2. **Clarify the sampling description in Section 4.1**
   Address Weakness (2) by explaining how identities or support samples are chosen, and how it affects the method’s generalizability.

3. **Further elaborate the motivation for introducing mAP**
   In response to Weakness (3), describe how mAP is computed during training or why it is crucial for the proposed method.

4. **Refine the pseudo-code training loop and forgetting mechanism**
   In response to Weakness (4), provide more information on the number of iterations for meta-loss training, and how exactly the MetaUnlearn_unlearning process is carried out (in batches, one by one, etc.). Including more detailed descriptions in the methods section would make it easier for readers to understand.

5. **Justify Formula (5) more thoroughly**
   In response to Weakness (5), explain the reasoning behind useing $\(L_{\text{task}}(D_f; \theta_u)\)$ not $\(L_{\text{task}}(D_v; \theta_u)\$).

**Strengths And Weaknesses:**

### Strengths

1. **Novel and practical problem setting**
   The paper focuses on a scenario where personal data must be deleted for compliance reasons, making it unavailable after training. This extreme yet realistic setup aligns well with real-world privacy concerns.

2. **A general, extensible framework**
   By adopting a meta-learning paradigm, the proposed method trains a meta-loss that can quickly adapt to unlearn specific data at inference time. This approach is naturally extensible to various tasks and model architectures.

3. **Comprehensive experiments**
   The authors evaluate their approach on CelebA and CelebA-HQ using multiple unlearning scales (i.e., varying numbers of identities). They measure both the trade-off between forgetting and retaining, as well as membership inference attacks (MIA). The results support the main claims.

4. **Comparison with multiple baselines**
   The paper compares MetaUnlearn with other unlearning algorithms (e.g., SSD, PGU, JiT) that do not require the original training set at inference time, thereby highlighting MetaUnlearn’s advantages.

---

### Weaknesses

1. **Insufficient ablation on the meta-loss importance**
   Although the meta-learning approach is shown to be effective, the paper does not thoroughly evaluate how much of the improvement is due to the learnable meta-loss $\(h_\phi(\cdot)\)$. An ablation study using a standard loss function instead of $ \(h_\phi(\cdot)\)$ would clarify its specific contribution.

2. **Unclear sampling procedure in Section 4.1**
I am a bit confused about whether identities  are sampled  or whether the support set \(S\) is  sampled. If \(S\) is sampled, how does this affect generalizability? For example, if a different forget subset is used, would the same mechanism remain valid? On the other hand, if identities are sampled, does this require access to all associated data, and how does this impact generalizability? Providing more details on the sampling strategy and its implications for generalizability would enhance the clarity of this section.


3. **Rationale and effect of the mAP need more elaboration**
While the paper mentions using mAP for alignment or scaling, the performance gain appears modest, and it is not clear how mAP is computed during training.

4. **Pseudo-code and meta-loss training procedure lack detail**
   Whether training the meta-loss parameters is done once or iteratively. Likewise, it is not specified whether the support samples in \(S\) are forgotten all at once or one by one. More details would make it easier for others to reproduce the results.

5. **Concern Regarding Formula (5)**
I would like to ask if
$\[
L_{\text{task}}(D_f; \theta_u) - L_{\text{task}}(D_v; \theta)
\]$
in the second term can be replaced by
$\[
L_{\text{task}}(D_v; \theta_u) - L_{\text{task}}(D_v; \theta).
\]$
If not, could you please provide an explanation?

---

> ### Author Response · Authors · 2025-01-11
>
> We thank the reviewer for his comments and appreciate that he recognized the novelty of our setting. We would like to address his concerns as follows:
> 1. We thank the reviewer for the suggestion. Replacing the meta-loss with another loss function (that does not require access to $D_r$) is equivalent to using existing methods [1, 2]. For instance, replacing $h_\phi(\cdot)$ with the loss proposed in [2] would be equivalent to just using PGU [2]. Therefore, ablating the method by replacing the loss function is equivalent to comparing JiT [1] and PGU [2] performance with ours. The comparison is already in the main paper in Tables 1 to 3 (Tables 1 to 4 in the revised version). On CelebA-HQ (Table 1), MetaUnlearn is always more effective than existing baselines, and, particularly efficient in the 50 identities unlearning case where it scores 95.9 of ToW compared to 94.9 and 59.2 of PGU and JiT. On CelebA (Table 2), MetaUnlearn performances are similar to PGU and JiT (e.g., 97.2 vs 96.9 and 97.6 of ToW in the 5 IDs case and 97.4 vs 97.2 and 96.9 of ToW in the 10 IDs case), however, in the average case, MetaUnlearn achieves better MIA scores (14.3% vs 15.8% of PGU, the second best). We improved the paper by clarifying this (Section 5.1).
> 2. Section 4.1 describes the sampling procedure used when training the meta-loss. As data must be stored for no longer than necessary, our meta-loss function must be trained after the original model training and before the data becomes unavailable. Therefore, we can iteratively sample random identities to form simulated unlearning requests as data is still available. After selecting random identities, we sample one image for each identity to build the simulated support set and, consequently, the simulated forget set. The simulated support set, the simulated forget set, and the validation set are used to optimize the meta-loss. We discard the simulated support and forget sets, and repeat this process until all identities are used to optimize the meta-loss at least three times (i.e. for three epochs, where an epoch consists of iterating once over all identities). We achieve generalizability by iterating over all training identities multiple times. We updated Sections 4.1, 4.2, and Appendix A to address the reviewer's concern, highlighting that our meta-loss is trained when the data is still available to the model owner. Additionally, we updated Figure 3, better visualizing the three steps of our method.
> 3. While the margin is modest, the accuracy term provides consistent improvements compared to not using it. Particularly, CelebA-HQ/50 got a ToW improvement of 0.5% (Table 6 of the revised paper) compared to not using it. Given its negligible computational cost, we decided to include it as part of our training pipeline.
> As per weakness 2, the model owner still has access to image labels while training the meta-loss, therefore, we can use those annotations to compute the mAP. From a technical standpoint, we adopted torchmetrics [3] implementation of the mean Average Precision for the actual calculation.
> 4. We thank the reviewer for this suggestion to improve our work. We partly answer the question about the meta-loss training in Weakness 2. About the unlearning step, all support samples are unlearned with a single gradient update both at training and unlearning time, without any iterative procedure. We included the list of hyperparameters used in Table 7a and a detailed description in Appendix A. Moreover, we updated Section 4 and Algorithm 1 of the revised paper to improve MetaUnlearn training explanation.
> 5. We thank the reviewer for the interesting question. The solution proposed by the reviewer is feasible and in practice may achieve competitive results (e.g., 97.1 vs our 97.2 ToW on CelebA/5, while 95.9 vs our 97.4 ToW on CelebA-HQ/20). However, this replacement requires storing the activations (for gradient computation) for two datasets, i.e., $D_f$ for the first term and $D_v$ for the second term. Differently, our method only stores the activations of $D_f$, with a much lower memory footprint (e.g., 20GB vs 12GB on CelebA/5). This is why we decided to use $\mathcal{L}_\text{task}(D_f; \theta_u)$ in both terms.
>
> [1] Foster, Jack, et al. "Zero-shot machine unlearning at scale via Lipschitz regularization." arXiv preprint arXiv:2402.01401 (2024).
>
> [2] Hoang, Tuan, et al. "Learn to unlearn for deep neural networks: Minimizing unlearning interference with gradient projection." Proceedings of the IEEE/CVF Winter Conference on Applications of Computer Vision. 2024.
>
> [3] https://lightning.ai/docs/torchmetrics/stable/detection/mean_average_precision.html

---

> > ### Comment · Reviewer_noQy · 2025-01-18
> > **Reply**
> >
> > Thank you for your response and revisions. Your explanations have addressed most of my concerns.

---

### Review · Reviewer_YfPq · 2025-01-02

**Summary Of Contributions:**

The paper proposes an unlearning algorithm that tries to remove the effect of unwanted data from the training set on a model without retraining it. Given some random subsets of training data, they propose to unlearn those sets and output a new $\theta_u\$ parameters of the model. They also propose an evaluation benchmark called 1-SHUI that is meant to compare unlearning methods in the absence of full training data.

**Audience:**

Yes

**Claims And Evidence:**

Yes

**Requested Changes:**

Please address the issues in the **Weaknesses** section.

**Strengths And Weaknesses:**

**Strengths**

The paper proposes an unlearning method called MetaUnlearn that removes the influence of some sets from the model. They construct a benchmark to evaluate the performance. The paper also lists the privacy issues of holding data after training, which can be problematic in most real-world settings.


**Weaknesses**
1. I believe this is meant to be learning-to-unlearn in the introduction section:
"Since unlearning using a single image per identity is challenging, we propose a learning-to-learn framework,
MetaUnlearn, ..."

2. This contribution term is misleading: "We propose 1-SHUI the first benchmark that evaluates machine unlearning methods in data absence." As far as I understood the evaluation, you still need the training set when creating the benchmark.

3. More explanation regarding this is also necessary: "We evaluate existing applicable MU methods on 1-SHUI, revealing that they struggle to tackle machine unlearning when training data is unavailable." Based on your results, some of the baselines, like PGU, perform very similarly to the proposed method and better in some cases.

4. The paper uses ToW metric as the main metric instead of the accuracies of the sets. I don't believe this is the best metric here as it may punish positive improvements compared to $\theta_r$ because of the absolute value. For example, in Table 2 (5 identities), PGU has 84.4, 82.1, and 80.9 mAP on the retain, forget, and test sets, respectively, while MetaUnlearn has 84.4, 81.9, and 80.9. On average, PGU is better but in ToW MetaUnlearn is better. The same goes for the JiT model.

5. Some results seem to have a high standard deviation. How many times were the results computed to get the mean and std?

In general, the paper is not well written, and it could be improved to make it more understandable and have fewer errors. The results have also shown very marginal improvements. I will update the claims and evidence part based on your response.

---

> ### Author Response · Authors · 2025-01-11
>
> We thank the reviewer for his comments and appreciate that he recognized the real and practical relevance of the problem. We would like to address his concerns as follows:
> 1. We thank the reviewer for the suggestion. We used the term as a synonym for “meta-learning” meaning that we are proposing a machine unlearning approach that leverages meta-learning principles. We updated the paper following the reviewer's recommendation.
> 2. We want to clarify that our contribution is a new benchmark that allows for identity unlearning in data absence. Of course, measuring unlearning effectiveness requires accessing the data and the retrained model, and, to the best of our knowledge, no approach enables such evaluation. In the referred sentence, we meant that we are the first to propose a benchmark for unlearning random identities without accessing original data. We revised the second contribution in the introduction, clarifying this aspect.
> 3. The proposed benchmark (1-SHUI) is an extremely challenging task that involves unlearning identities by relying on a single image. Although PGU performs similarly to MetaUnlearn in some cases, it is less reliable, as it is inconsistent across different datasets. For instance, when unlearning 50 identities on CelebA-HQ (Table 1 of the revised paper), PGU achieves 1 point less in ToW compared to MetaUnlearn, and it is less effective in protecting privacy under membership inference attacks. Similarly, on CelebA/10 (Table 2 of the revised paper), the membership inference attack can spot 6% more members (at 0.01% FPR) from a model unlearned with PGU than a model unlearned with MetaUnlearn. In CelebA/1 (Table 4 of the revised paper), although PGU achieves a slightly better ToW (+0.1 points) over MetaUnlearn, it fails to protect 0.26% of unlearned examples (at 0.01% FPR) from membership inference attacks. Instead, a model unlearned with MetaUnlearn completely protects forget data, scoring 0.01%TPR @ 0.01% FPR. Therefore, we believe MetaUnlearn still provides a promising direction for solving this challenging task.
> 4. We want to clarify a potential misunderstanding. To recap, the goal of machine unlearning is to update the model in a way such that it behaves as if it has never seen the data we aim to forget (or the forget set). In other words, the target for unlearning is approximating the retrained model output distribution [1], i.e., the model trained from scratch without the forget set. To this end, all unlearning algorithms are evaluated by measuring the discrepancy of performance between the unlearned model and retrained model. The rationale is that the unlearned model should behave like a retrained model in order to minimize information leakage about the forget set [1, 3]. Thus, like the majority of previous works [1, 2, 3, 4, 5], we measure unlearning performances based on the discrepancy with the retrained model, penalizing any misalignment. For example, if the retrained model achieves 79.1 mAP on the forget set, then the ideal unlearned model should get the same mAP. Therefore, contrary to standard learning scenarios, where a higher accuracy translates to a better model, in machine unlearning a higher forget mAP (or accuracy) does not imply a better unlearning algorithm.
> So, by looking at Table 2 (the 5 identities case) MetaUnlearn achieves a smaller forget mAP discrepancy (+2.9) with the retrained model compared to PGU (+3.1). We decided to rely on ToW [1] as it conveniently summarizes this discrepancy in a single value. Furthermore, we do not limit our evaluation to ToW and exploit the current SOTA in low-cost membership inference attacks RMIA [6], published in ICML 2024, which provides an extra important layer in evaluating methods protection from MIAs [1, 4]. Considering prior works [1, 2, 3, 4], we believe these metrics are comprehensive enough to compare our approach with existing baselines.
> 5. We ran each experiment three times with three different seeds (0, 1, 2) as in [3]. As the forget set size is relatively small compared to the other two sets (retain and test), there can be high forget accuracy fluctuations based on the classification difficulty, also observed in [3]. However, for ToW, the best methods achieve a low std as they align well with the retrained model.
>
> [1] Zhao, Kairan, et al. "What makes unlearning hard and what to do about it." In NeurIPS, 2024.
>
> [2] Liu, Jiancheng, et al. "Model sparsity can simplify machine unlearning." In NeurIPS, 2023.
>
> [3] Kurmanji, Meghdad, et al. "Towards unbounded machine unlearning." In NeurIPS, 2023.
>
> [4] Fan, Chongyu, et al. "Salun: Empowering machine unlearning via gradient-based weight saliency in both image classification and generation." In ICLR, 2024.
>
> [5] Maini, Pratyush, et al. "Tofu: A task of fictitious unlearning for llms." arXiv preprint arXiv:2401.06121 (2024).
>
> [6] Zarifzadeh, Sajjad, Philippe Cheng-Jie Marc Liu, and Reza Shokri. "Low-cost high-power membership inference by boosting relativity." In ICML, 2024.

---

> > ### Author Response · Authors · 2025-01-11
> >
> > We acknowledge the reviewer's consideration and hope our answers have clarified potential misunderstandings. We did a deep proofreading and improved the manuscript (see the revised version), as we are eager to enhance the quality of our work in order to make it more readable and to resolve possible errors. We also invite the reviewer to hint at the errors found and unclear sentences, so to help us further improve our paper by avoiding mistakes and potential confusion.

---

> ### Comment · Reviewer_YfPq · 2025-01-11
> **Reply**
>
> The authors have made considerable changes to clarify their contribution and avoid misleading parts in the benchmark vs. the unlearning method. The main figure is also much better than it was initially. They have also added better explanations at different sections. I would advise the authors to do the same in the conclusion section. The improvement of the MetaUnlearn method is still very minimal, and with high standard deviation, it's very hard to quantify that their method is clearly better than the baselines. Nonetheless, considering that they are proposing both a benchmark and an unlearning method, I believe the research work might be important and have updated the Claims and Evidence part.

---

> > ### Author Response · Authors · 2025-01-16
> >
> > We thank the reviewer for the positive feedback. Following the reviewer's suggestion we updated the conclusions section (Section 6) by clarifying the proposed task and MetaUnlearn's training pipeline. We are open to adding further clarifications if needed.

---

### Author Response · Authors · 2025-01-11

We thank the reviewers for their positive feedback, recognizing the importance of the problem (Hsay, noQy, YfPq), the extensibility of the framework (noQy), the method effectiveness (Hsay), and the comprehensiveness of the experiments (Hsay, noQy). We appreciate the suggestions and updated the manuscript by:
1. Improving Section 1 to emphasize the limitation of previous approaches when unlearning in data absence scenarios.
2. Enhancing Section 4 to account for confusion about data availability and MetaUnlearn training.
3. Improving Algorithm 1 to highlight each step in MetaUnlearn optimization.
4. Revising Figure 3 to emphasize different MetaUnlearn steps and the data availability on each step.
5. Comparing methods on a new dataset to show that MetaUnlearn is not specifically designed for face attribute recognition.
6. Expanding Appendix A to further describe the training details of MetaUnlearn.

Below we answer each concern in detail.

---

### Decision · Action_Editor_6gXb · 2025-02-08

**Recommendation:** Accept as is

**Comment:**

Before rebuttal, all reviewers were positive about the submission, though they raised concerns regarding the clarification of contributions and technology, as well as evaluation. The authors addressed these concerns. All reviewers agreed that the proposed method is effective, with solid benchmark contributions, and recommended acceptance. Thus, the AE also recommends accepting the paper as is.

However, one reviewer does not recommend transferring this paper from ICLR journal track to conference track, reasoning that the proposed unlearning algorithm shows only minimal improvement. The AE concurs with this comment.

**Audience:**

Yes

**Claims And Evidence:**

Yes